# Planktonic foraminifera-derived environmental DNA extracted from abyssal sediments preserves patterns of plankton macroecology

Raphael Morard[1], Franck Lejzerowicz[2], Kate F. Darling[3,4], Béatrice Lecroq-Bennet[5], Mikkel Winther Pedersen[6], Ludovic Orlando[6], Jan Pawlowski[2], Stefan Mulitza[1], Colomban de Vargas[7,8] and Michal Kucera[1]

[1]*MARUM Center for Marine Environmental Sciences, University of Bremen, Leobener Strasse, 28359 Bremen, Germany*

[2]*Department of Genetics and Evolution, University of Geneva, Geneva, CH1211, Switzerland*

[3]*School of GeoSciences, University of Edinburgh, Edinburgh EH9 3FE, UK*

[4]*School of Geography and GeoSciences, University of St Andrews, Fife KY16 9AL, UK*

[5]*Okinawa Institute of Sciences and Technology Graduate University, 1919-1 Tancha, Onna-son 904-0495, Japan*

[6]*Centre for GeoGenetics, Natural History Museum of Denmark, University of Copenhagen, Øster Voldgade 5-7, Copenhagen 1350K, Denmark*

[7]*Centre National de la Recherche Scientifique, UMR 7144, EPEP, Station Biologique de Roscoff, 29680 Roscoff, France*

[8]*Sorbonne Universités, UPMC Univ Paris 06, UMR 7144, Station Biologique de Roscoff, 29680 Roscoff, France*

**Keywords:** Planktonic foraminifera, metabarcoding, DNA preservation, eDNA, protists.

*Corresponding author: Raphaël Morard, MARUM Center for Marine Environmental Sciences, University of Bremen, Leobener Strasse, 28359 Bremen, Germany,Phone: +49 (0) 421 218 -65973, Fax: +49 (0) 421 218 – 9865974, email: rmorard@marum.de.*

**Abstract**

Deep-sea sediments constitute a unique archive of ocean change, fueled by a permanent rain of mineral and organic remains from the surface ocean. Until now, paleo-ecological analyses of this archive have been mostly based on information from taxa leaving fossils. In theory, environmental DNA (*e*DNA) in the sediment has the potential to provide information on non-fossilized taxa, allowing more comprehensive interpretations of the fossil record. Yet, the process controlling the transport and deposition of *e*DNA onto the sediment and the extent to which it preserves the features of past oceanic biota remains unknown. Planktonic foraminifera are the ideal taxa to allow an assessment of the *e*DNA signal modification during deposition because their fossils are well preserved in the sediment and their morphological taxonomy is documented by DNA barcodes. Specifically, we re-analyze foraminiferal-specific metabarcodes from 31 deep-sea sediment samples, which were shown to contain a small fraction of sequences from planktonic foraminifera. We confirm that the largest portion of the metabarcode originates from benthic bottom-dwelling foraminifera, representing the *in-situ* community, but a small portion (< 10%) of the metabarcodes can be unambiguously assigned to planktonic taxa. These organisms live exclusively in the surface ocean and the recovered barcodes thus represent an allochthonous component deposited with the rain of organic remains from the surface ocean. We take advantage of the planktonic foraminifera portion of the metabarcodes to establish to what extent the structure of the surface ocean biota is preserved in sedimentary *e*DNA. We show that planktonic foraminifera DNA is preserved in a range of marine sediment types, the composition of the recovered *e*DNA metabarcode is replicable and that both the similarity structure and the diversity pattern are preserved. Our results suggest that sedimentary *e*DNA could preserve the ecological structure of the entire pelagic community, including non-fossilized taxa, thus opening new avenues for paleoceanographic and paleoecological studies.

## 1 Introduction

With over two thirds of the planet covered by oceans, deep-sea deposits form the most extensive archive of the Earth's recent history. These deposits preserve mineralized skeletons of marine nano- and micro-plankton, which serve as an record of past climate (e. g., Hillaire-Marcel and de Vernal, 2007) and biodiversity (Yasuhara et al., 2015) changes. However, planktonic groups leaving fossilized remains only represent a small fraction of the marine diversity (de Vargas et al., 2015). In theory, environmental DNA (*e*DNA) buried in marine sediments can provide information on the history of marine organisms that do not produce fossils (Pedersen et al., 2015). Deep sea sediments are rich in

DNA with $0.31 \pm 0.18$ g of DNA per m² in the surface layer, and more than 90% of this DNA is extracellular (Dell'Anno and Danovaro, 2005). This means that DNA from many organisms is preserved after their death in the sediment and the high abundance of the DNA indicate that at least a part of the DNA pool derives from organisms living in the water column above the sediment (Lejzerowicz et al., 2013). Part of this DNA pool remains preserved in ancient sediments, and can be extracted and analyzed using metabarcoding to reveal the molecular diversity of past ecosystems (Lejzerowicz et al., 2013; Pawłowska et al., 2014). This potential has been demonstrated in a range of other depositional environments, such as cave sediments, lake and ice cores where the dynamics of plant and animal communities could be followed over 50 ka (Pedersen et al., 2015).

In marine sediments, the presence of *e*DNA sequences has been reported from organic-rich layers in the Mediterranean dating back to 217 ka (Coolen and Overmann, 2007) and 125 ka (Boere et al., 2011), in sediments covering the last 11.4 ka in the Black Sea (Coolen et al., 2013), and in up to 32.5 ka old deposits in the Atlantic (Lejzerowicz et al., 2013; Pawłowska et al., 2014). Recently, Kirkpatrick et al. (2016) showed that the abundance of planktonic DNA was decreasing within 100-200 ka in sediments of the Bering sea but traces were still detected in sediments up to 1.4 Ma. Direct comparison with co-occurring fossils showed that the sequenced *e*DNA pool exceeds the taxonomic spectrum of the fossils, but many of the taxa preserved as fossils were not identified in the *e*DNA (Pawłowska et al., 2014; Pedersen et al., 2013). This raises the question of how well the sedimentary DNA pool reflects the autochthonous (in situ origin) or allochthonous (external origin) community composition, whether there is any differential DNA preservation across taxa and whether the metabarcode marker selected is fully representative of the entire taxonomical diversity, regardless of its origin. The extensive fragmentation of *e*DNA (Pedersen et al., 2015) makes incompatible the amplification of sequences longer than ~100 bp, preventing the access to long and informative barcodes.

The primary difficulty in the analysis of the sedimentary DNA pool is to separate the local and allochtonous origin of the sequenced material (Torti et al., 2015). This can be done with certainty only when the ecological origin of the sequenced eDNA is unambiguously resolved. Potential bias could arise from a range of factors including preferential amplification (Taberlet et al., 2012), inconsistent taxonomic resolution of the sequenced barcodes (Pawlowski et al., 2012) and insufficient coverage of the barcode reference database (Pawlowski et al., 2014b).

Here we take advantage of the possibility to unambiguously ascribe sequences of foraminifera to benthic and planktonic lineages. By analyzing the planktonic portion of foraminiferal metabarcodes from deep sea sediments, we provide evidence that the structure and diversity of surface ocean communities is preserved in *e*DNA molecules and that the preservation is not limited to specific depositional environments. We focus our analysis on the Foraminifera because of access to highly resolving short barcodes (Pawlowski and Lecroq, 2010) and the availability of a taxonomically well resolved barcode database for the planktonic taxa (Morard et al., 2015). It allows the unambiguous separation of the benthic, autochthonous, component of the dataset from its planktonic, allochthonous, component.

Foraminifera are single-cell eukaryotes (protists) belonging to the phylum Rhizaria (Adl et al., 2012). Most Foraminifera lineages occupy benthic ecological niches. Their ~ 5000 morphospecies inhabit the bottom of shallow coastal environment to deep abyssal plains. In contrast, the planktonic lineages only include 50 morphospecies, living mostly in the photic part of the water column. They are found from tropical to polar water masses and spend their entire life cycle in the plankton (Hemleben et al., 1989). After their death, planktonic foraminifera sink to the bottom of the ocean where they are found in the calcareous ooze, ranging from ~1 to 4,5 km water depth and distributed from low to high latitude (Dutkiewicz et al., 2016). The fossil planktonic assemblages are preserved without taxonomic bias above the lysocline and become increasingly affected by the preferential dissolution of thin-shell species below this limit (Berger and Parker, 1970). Foraminifera are known for their unusually high rate of evolution (de Vargas et al., 1997) resulting in highly resolving barcodes even in fragments shorter than ~100 bp thus allowing unambiguous species identification with relatively short barcodes (Pawlowski and Lecroq, 2010). In addition planktonic foraminifera harbor considerable cryptic diversity (Darling and Wade, 2008; Morard et al., 2016), which offers an additional layer of taxonomic information that can be exploited in eDNA studies. Therefore, planktonic foraminifera possess barcodes with resolution that is equal or higher than their benthic counterparts. This facilitates the taxonomic identification of short, potentially degraded, *e*DNA sequences.

In the present study we perform new analysis on *e*DNA libraries generated by Lecroq *et al.* (2011), which comprise metabarcodes from 31 abyssal sediment samples containing ~ 78 million foraminiferal sequences derived from the 37f foraminiferal specific barcode of the 18S rDNA. The major portion (>99%) of the sequences could be assigned to benthic taxa and their composition was analyzed to unravel the patterns of benthic diversity on the sea floor. However,

a tiny portion of the barcodes (<1 %) could be assigned to planktonic foraminifera. These sequences represent eDNA exported to the seafloor from the plankton. With the recent development of the *Planktonic Foraminifera Ribosomal Reference* Database (PFR², (Morard et al., 2015)), the environmental sequences belonging to planktonic foraminifera in the *e*DNA libraries generated by Lecroq *et al.* (2011) can now be for the first time thoroughly analyzed and assigned

5    to the morphological and cryptic species levels

The extensive knowledge on the distribution and abundance of planktonic foraminiferal shells in surface sediments (Kucera et al., 2005), enabled the *e*DNA data to be directly compared with data derived from classical taxonomy. We thus assess to what extent the eDNA originating from plankton is representative of the source community which is an essential prerequisite for interpretation of the eDNA archive in the sediment.

**2 Material and Methods**

The 31 surface sediment samples analyzed were taken at water depths ranging from 1,745 to 5,338 m and cover sediment types from calcareous ooze in the Caribbean Sea to fine clastic sediments in the Arctic Ocean (Fig. 1a, Supplement 1). All analyses are based on the Illumina Solexa GAII datasets generated by Lecroq et al. (2011) and

registered at the NCBI's Short Read Archive under the BioProject number PRJEB2682. The original sequencing data include 78,613,888 reads covering the 36 positions starting 3' of the "GACAG" motif delimiting the foraminifera-specific hypervariable region 37f region (Pawlowski and Lecroq, 2010).

We used the unique sequences obtained for each library in Lecroq et al. (2011) after the strict dereplication step and the removal of singletons associated with only one read occurrence in a library. For each DNA library, we parsed

sequencing reads passing the default base calling of GAPipeline v 1.0 and reads showing a single base quality or averaged base qualities inferior to 10 and 20, respectively as well as sequencing reads presenting ambiguities (N) or homopolymers over 30 positions. This resulted in a total of 204,704 unique and filtered 36 bp-long sequences representing 39,210,426 reads (Supplement 1, Fig 1b). During the generation of the data, one sample was used as a control to check for potential cross-contamination. This sample consisted in the DNA extract of a single cultured

species: *Reticulomyxa filosa*. The sequencing of this sample produced 2,416,756 reads, corresponding to 1,689 dereplicated tags with at least 2 reads per tag. After filtering and clustering, we recovered only one OTU, which was

identical to the 37f hypervariable sequence of *R. filosa* previously obtained by using classical Sanger technology, thus showing the absence of cross contamination (Lecroq et al., 2011).

We compared the retained reads to the *Planktonic Foraminifera Ribosomal Reference* database (PFR², (Morard et al., 2015)), which represents a compilation of 3,322 curated partial SSU rDNA sequences of planktonic foraminifera groups associated with a 6-ranks taxonomy. The ranks reflect taxonomic units and are organized into a hierarchal framework, with the basal ranks being the coarsest units and the terminal ranks corresponding the finest taxonomic levels. The first three basal ranks correspond to the level of assignation comparable to that achievable using morphological data, and is thus analogous to fossil data. The three terminal levels correspond to the molecular taxonomy accessible using molecular data only. The PFR² taxonomic framework derives from single-cell genetic studies where the molecular taxonomy (definition of genetic types) was based on phylogenetic inferences and or/automatic delimitation methods. The delimited cluster of sequences were then compared to ecological and biogeographical data to validate their status as genuine biological species (see Morard et al., 2015 and references herein). Of the 3,322 sequences available in PFR², 2,418 sequences covered the fragment of the region 37f. These sequences were downloaded from the PFR$^2$ database (http://pfr2.sb-roscoff.fr/) and trimmed to the 36-nt fragment corresponding to the environmental sequences, which resulted in a total of 463 unique homologous reference sequences (Supplement 2). Initially, we evaluated the taxonomic resolution of the 36 nt barcoding region and found that it was variable enough to discriminate the genetic types (equivalent to cryptic species) within morphological species of almost all planktonic foraminifera taxa referenced in *PFR$^2$*. We observed a lack of genetic resolution (different taxonomic entities yielding identical barcodes) for only two species pairs belonging to *Globorotalia* (*tumida* and *ungulata*) and *Globigerinella* (*calida* and *siphonifera*) and three pairs of genetic types among *Globorotalia truncatulinoides* (type III and IV), *Pulleniatina obliquiloculata* (types I and II) and *Globigerinita glutinata* (types III and IV).

We then individually aligned the 4,466 to 27,578 unique sequences obtained for each of the 31 samples against the 461 reference sequences using the Needleman-Wunsch global sequence alignment algorithm (Needleman and Wunsch, 1970), to separate the portion of the dataset belonging to the planktonic foraminifera (allochthonous origin) from the portion belonging to the benthic foraminifera (autochthonous origin). Pairwise genetic distances were

calculated as the number of differences (counting successive indels and terminal gaps as one difference), and an iterative clustering of the unique environmental sequences with the reference sequences was performed, allowing 1, 2, 3, 5 and 10 differences as thresholds for the average linkage algorithm. We then extracted all environmental sequences found within each cluster containing a planktonic reference sequence in an iterative manner, by screening from the most stringent (1 difference threshold) to the most permissive (10 differences threshold) clusters. As a post hoc verification, we compared these sequences with the extensive benthic foraminifera sequence database used in Pawlowski *et al.* (2014a) together with the sequences of the *Protist Ribosomal Reference Database* (*PR²*, Version based on Release 203 of Genbank, (Guillou et al., 2013)) and additional undescribed benthic specimen sequences to ensure that the extracted sequences do not belong to benthic foraminifera. No match was found. We assigned to each extracted environmental sequence the taxonomy of the planktonic reference sequences in the cluster. Finally, we retained only the sequences occurring in at least 2 samples or having a minimal abundance of 10 for downstream analysis. The final product was then considered an individual e-ribotype (Supplement 3). E-ribotypes are unique environmental sequences (not cluster) originating from planktonic foraminifera and thus transferred from surface ocean to the bottom (allochthonous origin). The relative proportions between e-ribotypes (planktonic reads) and the benthic reads of each sample are shown on Fig. 1B. We calculated the rarefaction curves of each individual samples using PAST v 2.17 (Hammer et al., 2001) to estimate to what degree the full taxonomic spectrum of each sample was recovered by *e*DNA (Fig. 2).

Genuine sequences of planktonic foraminifera representing species not yet registered in the reference database may have been omitted. We therefore structured our analyses to account for the detection of possibly unknown genetic types. To this end, we used the phylogenetic signal contained in the 36bp-reads to build a taxonomic framework within each morphospecies. In contrast to strict annotation approaches using arbitrary similarity thresholds, a phylogenetic approach can identify novel genetic type, not represented in the reference comparative database. The retained e-ribotypes were automatically aligned using MAFFT v.7 (Katoh and Standley, 2013), with reference sequences of the complete 37f region. The complete 37f region was used at this step instead of the 36-bp fragment to avoid possible read alignment shifts caused by artificial mismatches with trimmed 36-bp sequences during the assignment process. A single alignment was produced per morphospecies. For each resulting alignment, a phylogenetic tree was inferred using PhyML (Guindon et al., 2010) implemented in SEAVIEW 4 (Gouy et al., 2010) with default option using aLRT for branch support estimation. The resulting trees were visualized with ITOL (Letunic and Bork, 2011) and all visually

distinct clusters were considered as unique genotypes (Supplement 4). The reads clustering with reference sequences were assigned at the genetic type level, the sequences clustering without a close reference received an artificial genetic type attribution (Supplement 3). These assignments were used to prepare three datasets with different degrees of taxonomic resolution (at the level of e-ribotype, genetic types and morphological species). The occurrences of the defined genetic types in the samples are shown on the Figure 3.

The difference in amount of reads recovered between libraries was normalized using the Cumulative Sum Scaling method (Paulson et al., 2013) available on the metagenomeSeq Bioconductor package (Paulson et al. 2016) in R (R Development Core Team, 2014). The Cumulative Sum Scaling corrects the biases induced by differential sequencing depths and uses a Zero-Inflated Gaussian distribution mixture model that accounts for technical zero value resulting from under-sampling. The taxonomic richness and structure of the normalized datasets for each taxonomic resolution level were analyzed using Non-Metric Distribution Scaling (NMDS) as implemented in PAST v 2.17 (Hammer et al., 2001), associated with one-way PERMANOVA to test for significance of distribution difference between groups (Table 1). We used the Dice distance to consider only the presence/absence data and the Bray-Curtis distance to compare absolute and relative abundances of reads among the samples (Fig. 4). To compare the similarity structure and diversity in the samples based on the *e*DNA reads with census counts of microfossils, we used the MARGO database. The census count represent the relative abundance of species observed in a fossil assemblage based on the count of typically 300-500 specimens (Kucera et al., 2005). We calculated the fossils-based diversity (Shannon-Wiener) and similarity (Dice and Bray-Curtis) matrices using PAST v 2.17 for all surface samples within the regions outlined in Fig 1A (between 6 and 13 per region, Fig. 5 and 6).

**3 Results**

After quality filtering and collapsing of identical reads into single sequences, the comparison of the entire dataset with reference databases (Supplement 2) allowed to ascribe with certainty 1,373 unique sequence patterns representing 488,291 reads to planktonic foraminifera (Supplement 1, 3). Because we required reads to be present in a minimum of two samples or to show a minimal abundance of 10 in the entire dataset, the retained dataset was reduced to 697 unique sequences of planktonic foraminifera (e-ribotypes), which are representing a total of 486,435 reads (~0.63%

of the total dataset, Supplement 1). Diversity was then assessed using a phylogenetic approach and the 697 e-ribotypes were found to represent 37 genotypes (Fig. 3, Supplement 4). Of these, 675 e-ribotypes (representing ~ 99 % of the planktonic reads) were attributed to 24 genotypes already detected in plankton and assigned to 17 morphological species (Supplement 3, 4). The remaining 22 e-ribotypes clustered into 13 genotypes with no apparent affinities with

the genotypes detected in plankton. These e-ribotypes represent only ~0.5% of the planktonic reads.

After this filtering, between 48 (Library #SFA-17) and 124,355 (Library #SFA-15) reads were retained in 28 samples (Supplement 1, Fig. 1, 2), representing between 0.003 and 9.412% of the total foraminifera reads in the libraries from these samples (Fig. 1c). Three Arctic samples did not yield any sequences that could be assigned to planktonic foraminifera (Fig. 1d, 3). The total number of reads per sample is a function of sequencing effort and is therefore not

related to initial community density. However, the relative abundance of reads assigned to planktonic foraminifera in the DNA accumulated on the sea floor should reflect the relative proportion of the foraminiferal DNA produced by planktonic communities and the DNA produced by the in-situ benthic community. Whilst the absolute number of planktonic reads varied among the samples and replicates (Fig. 1d), we did indeed observe a higher reproducibility of the relative number of planktonic reads recovered from replicates at the same location (Fig. 1e). The relative

abundance of planktonic reads seems unrelated to the latitude or depth of the sample location (Fig. 1c, e). The samples with the highest relative proportions originate from Japan (0.790% to 9,412%) whilst the lowest abundances are observed in the Caribbean samples (0.003 to 0.032%). The high latitude samples (Arctic, North Atlantic and South Atlantic) show relative abundances ranging from 0.011 to 1.204% when excluding samples without planktonic reads. The relation between water depth and relative sequence abundance is not clear (Fig. 1b). It does not seem that the

proportion of planktonic DNA reads decreases with increasing depth, suggesting that bentho-pelagic flux exporting planktonic DNA do not weakens compared to the *in situ* community. Rarefaction analysis has been used to assess the degree to which the retained planktonic reads cover the diversity they contain (Fig. 2). As expected, the general trend indicates a higher degree of saturation in samples with more reads. For example, samples with the highest number of reads (Japan) had saturated diversity (Fig. 2a), whereas the Caribbean samples representing a similar geographical

province but with fewer reads are clearly under-saturated (Fig. 2c). However, we observe that samples from high latitude regions are also saturated (Fig 2b, c), despite having a lower number of reads than the samples from Japan, implying lower diversity.

With respect to the composition of the reads, we observed that e-ribotypes attributed to the microperforate species *Globigerinita glutinata* dominated the dataset (~77% of the reads) and were particularly abundant in subtropical communities (Fig. 3). E-ribotypes of common subtropical species *Orbulina universa*, *Globorotalia menardii, Globorotalia hirsuta*, *Hastigerina pelagica Neogloquadrina dutertrei*, *Globigerina falconensis*, *Globigerinella siphonifera, Pulleniatina obliquiloculata, Galitellia vivans* and *Candeina nitida* were found in subtropical samples, whereas e-ribotypes assigned to the species *Globigerinita uvula* and *Neogloboquadrina pachyderma* appeared to dominate subpolar and polar samples. Among these, e-ribotypes belonging to the genotype IV of *N. pachyderma* were mostly found in the Southern Ocean (>99.99%) whereas e-ribotypes of the genotype I were only observed in the subpolar samples from the northern hemisphere. Additionally, different *Globigerina bulloides* e-ribotypes were detected either in subtropical samples (type I) or in subpolar assemblages (type II). Similarly, type II e-ribotypes of *Globigerinita uvula* were found more frequently in subpolar samples from both hemispheres, whereas e-ribotypes of type I were also abundant in low-latitude samples (Fig. 3).

Prior to analyses of diversity patterns, we used the Cumulative Sum Scaling (Paulson et al., 2013) to correct for potential technical zero (i. e. undetected taxa due to undersampling) and for biases in relative proportions of taxa at the level of morphological species, genotypes and e-ribotypes. We calculated similarity matrices among samples using the corrected reads abundances at the three taxonomical levels in order to identify patterns of community structure. Visualization was based on Nonlinear Multi-Dimensional Scaling (NMDS, Fig. 4) with either Dice (Presence/absence) or Bray-Curtis similarity metrics computed from relative as well as absolute read abundances. Calculation performed on absolute number with Bray-Curtis (Fig 4a-c) showed high reproducibility for samples from the same regions, best expressed at the e-ribotype taxonomic level (Fig 4a). The high latitude communities are more similar at morphospecies level (Fig 4c), than at genotype level (Fig 4b). The Caribbean and Japan samples are closer at these taxonomic levels and even partly superposed at the genotype level. When relative proportions are considered (Fig 4d-f), the samples of the Caribbean and Japan region cannot be distinguished (Table 1)showing that the relative proportions of the major taxa are the same between these regions (Fig. 3). We also observe a clear separation between low and high latitude samples (Fig 4f) at morphospecies level, which is analogous to the structure represented by fossil assemblages in nearby samples (Fig 4g). Calculation performed on Dice indices (Fig 4i-k) tends to reproduce the same structure as the calculation performed with Bray-Curtis calculated on absolute read number (Fig 4a-c), especially at e-ribotypes level (Fig 4i), but the patterns are noisier. This is most likely due to the different level of taxonomic saturation between

the samples (Fig. 2). Remarkably, despite the different numbers of reads and the associated different level of taxonomic saturation, the recovered pattern of taxonomic composition of the reads is so strong that the opposition between the high and low latitude samples, clearly observed with fossil assemblages, remains even in the eDNA assemblages when considering the morphospecies level (Fig 4k-l). The signal in the eDNA data is noisier because only a fraction of the morphospecies have been detected by the eDNA (1 to 11 morphospecies per sample, Supplement 1), whilst 1 to 24 morphospecies are observed in the census counts. This means that the relative proportions of the reads carry enough information to reproduce similar patterns between eDNA and fossil record despite only a partial coverage of the morphological diversity (Fig 4f-g).

These observations taken together imply that the relative abundance of planktonic eDNA reads in the sediment samples contains exploitable information at all three taxonomic (morphological species, genotypes and e-ribotypes) levels. To further explore the diversity patterns implied by eDNA data, we calculated the Shannon-Wiener diversity index within each sample (Fig. 5a). Despite differences in sediment type and sequencing depth, eDNA in the analyzed samples reproduces the latitudinal diversity gradient based on morphospecies abundances in surface sediment samples (Rutherford et al., 1999). The latitudinal diversity gradient is present at all three taxonomic levels, but is most pronounced at the e-ribotype level (Fig. 5b).

Finally, since census counts of planktonic foraminifera morphospecies in surface sediments are available from the same regions as those analyzed for *e*DNA (Fig. 1a), we assessed whether the e-ribotype abundances reflect the same community turnover pattern as that indicated by fossil assemblages (Fig. 6). To this end, we compared pairwise distances between *e*DNA MOTU assemblages with pairwise distances between fossil assemblages. This comparison reveals that *e*DNA and morphospecies community turnover rates are significantly correlated (Fig. 6), with highest similarity among samples from the same region and lowest similarity among samples from different climatic regimes. This pattern emerges both when relative abundances and presence/absence data are considered. This implies that the proportionality of *e*DNA reads abundance is consistently scaled with the proportionality of plankton flux to the seafloor. The analysis based on relative abundances yields a pattern with highly consistent results for comparisons between climatic zones and more scatter when comparing samples within a region or within one climatic zone. This is likely due to the fact that the eDNA data only cover a part of the morphological diversity of the foraminifera

combined with differential distortion of the original abundance signal due to variation in gene copy number (Weber and Pawlowski, 2013) and primer bias (Bradley et al., 2016).

**4 Discussion**

Here, we provide evidence that *e*DNA originating from planktonic foraminifera is indeed preserved in the DNA pool of abyssal marine sediments irrespective of water depth, geographic region and sediment type. Earlier *e*DNA studies on marine sediments assumed that DNA preservation is proportional to the preservation of organic matter and, thus, prioritized sampling in organic-rich sediment layers (Coolen et al., 2009). Yet, recent experimental research and field studies suggest that the primary structure of DNA molecules is adsorbed to solid particles and molecules preserved in this way may form an archive of extracellular DNA regardless to the organic content of the sediment (Corinaldesi et al., 2007, 2011, 2014; Torti et al., 2015). We also show that the *e*DNA composition consistently reflects the composition of the pelagic planktonic communities from which it was derived (Figs. 5, 6). The high reproducibility of reads diversity (Dice index) and relative abundance (Bray-Curtis index) within a single region (Fig. 4) suggests that the taphonomic process governing the transfer and preservation of extracellular DNA from surface to bottom ocean are similar at regional scale and do not differentially impact DNA from species within different ecological groups.

Although the number of planktonic foraminifera reads recovered differed by three orders of magnitude between the Caribbean (62 to 212 reads per samples, representing ~0.003 to 0.03 % of the dataset) and Japan (3,620 to 124,355 reads per sample, representing 0.8 to 9.4 % of the dataset), the information recovered was sufficient to unveil the structure of foraminifera communities across the whole range of environments investigated (Figs. 3-6). However, since the taxonomic richness in *e*DNA data increased with sequencing efforts (Fig. 2), the recovery of the full taxonomic diversity requires a certain minimum sequencing effort. From the analyzed dataset, it is not possible to explain the large variation in the numbers of reads ascribed to planktonic foraminifera among regions (Fig. 2). This could represent DNA differential preservation conditions, or an imbalance between flux from the surface, allochthonous community and the abundance of DNA from the benthic, autochthonous community. The latter is a likely explanation because the analyzed *e*DNA material was amplified by PCR primers annealing to all foraminiferal sequences (Lecroq et al., 2011). During the PCR, the DNA of planktonic foraminifera might well be outcompeted by

the autochthonous DNA of benthic foraminifera, which is potentially more abundant, less damaged and more easily extracted from cells than when tightly absorbed to sediment particles (Ceccherini et al., 2009; Torti et al., 2015). It is noteworthy that the relative proportion of sequence reads may reflect the relative proportion of DNA molecules – but not necessarily that of cells – as shown in the case of a mock foraminiferal DNA community amplified using foraminiferal-specific primers (Esling et al. 2015)

Consistent with earlier studies (Capo et al., 2015; Lejzerowicz et al., 2013; Pawłowska et al., 2014; Pedersen et al., 2013), the taxonomic diversity revealed by the analyzed *e*DNA barcodes overlaps only partly with the diversity based on fossils present in the sediment. One part of the observed difference could be ascribed to the limited coverage of the reference database. Because of the way we assigned reads to planktonic foraminifera, we cannot assess the portion of the planktonic foraminifera diversity not represented in the reference database, although all major planktonic foraminifera taxa making >90% of tests larger than 150 μm are present in the reference database (Morard et al., 2015). We note, however, that our method allowed the discovery of unknown e-ribotypes clustering within e-ribotypes of known morphological species. Despite the discovery of the new e-ribotypes, the vast majority (99 %) of the retained reads could be associated with known genetic types. This exemplifies that the overlap of the *e*DNA reads library is large for well-studied taxa.

However, there might be a PCR bias that impairs the detection of some species. Indeed, none of the recovered barcodes could be attributed to four common species in the fossil record and well represented in the reference database: *Globorotalia truncatulinoides*, *Turborotalita quinqueloba*, *Trilobatus sacculifer* and *Globigerinoides ruber*. This observation is consistent with preferential PCR amplification. The rDNA of planktonic foraminifera is characterized by high and variable substitution rates (de Vargas et al., 1997), and two of the four above species exhibit some of the highest mutation rates (Aurahs et al., 2009). The manual inspection of a multiple sequence alignment containing the reference database sequences (Morard et al., 2015) revealed the presence of up to 5 mismatches between these species sequences and the primer sequences used to generate the dataset. Hence, such mutations in the conserved regions of the gene where the primers anneal may be responsible for detection failures. Another preferential PCR amplification could also explain the strong skew dataset towards microperforate species sequences, which represent 55 to 99% of the reads (Fig. 3), but only 0 to 30% of the morphological assemblages (Kucera et al., 2005). The microperforate clade

appears to have significantly lower rDNA substitution rates (Aurahs et al., 2009) and here we observe no mismatch between the primer and the reference sequences within this clade.

Alternatively, the higher abundance of reads assigned to microperforate taxa could represent a genuine pattern, questioning the representativeness of census counts of fossil foraminifera, which ignore specimens smaller than 150 µm (Kucera et al., 2005). Microperforate species tend to be small and are disproportionately abundant in the size fraction smaller than 150 µm (Brummer et al., 1986). This is significant because the *e*DNA archive comprises information on all planktonic foraminifera irrespective of size and is thus potentially a more comprehensive recorder of species proportions in the plankton.

Overall, our results indicate PCR/primer bias as the important limitation of planktonic foraminiferal community surveys based on metabarcoding. Alleviating them will allow detection of the full taxonomic spectrum, provided that sufficient sequencing effort is achieved, as recently discussed for fungi (Adams et al., 2013a, 2013b). To our knowledge, the dataset we re-analyzed represents the largest sequencing data for a given taxonomic group. Yet, it seems to indicate that the main ecological pattern can be extracted even from metabarcodes found at relatively modest frequencies (< 1000 reads, Figs. 3, 4: Caribbean samples). This conclusion underlines the importance of comprehensive reference datasets and barcoding efforts to facilitate the development of specific and effective probing techniques to recover the signal of individual key groups (Pawlowski et al., 2012).

Metabarcoding surveys of marine sediments offer a powerful alternative to study marine plankton ecology and biogeography. Plankton *e*DNA diversity observed in sea floor sediments represents a continuous flux of biomass, averaged over seasons and throughout the entire water column. Unlike plankton sampling, sea-floor deposits are not affected by the seasonality, reproductive cycle or habitat depth of the plankton at the time of sampling. They offer a spatiotemporally archive of the overlying water column, which contains an integrated record of the maximum range of taxa that is realized at least at some point during the seasonal cycle. In this way, it is possible to constrain biogeographical patterns like endemism or ecological exclusion across oceanic gradients, without the need for highly time-resolved sampling. Importantly, *e*DNA data can be used to test the stability of biotic interactions inferred from the plankton (Lima-Mendez et al., 2015) simultaneously across a large range of environmental conditions represented in the sediment.

**5 Conclusion**

Assuming that *e*DNA deposited on the sea floor is also preserved through time, marine sediments should contain a remarkable ancient DNA (*a*DNA) archive of the history of the complete plankton communities. There is growing evidence that *e*DNA is preserved in marine sediments old enough to cover the previous ice age (Lejzerowicz et al., 2013). Until now, the interpretation of *a*DNA datasets from marine sediments suffered from insufficient sequencing depth (Coolen et al., 2009) or insufficient coverage of the reference database (Pawlowski et al., 2014a). As a result, to which degree the observed *a*DNA patterns reflect genuine past ecological shifts and community structure remained contentious. Indeed, an investigation of a lake environment showed that only 71% of the *e*DNA diversity identified in the water column was preserved in the sediments (Capo et al., 2015) and that the DNA from taxonomic groups with fragile cell membrane such as Haptophya or Cryptophyta was less preserved in sedimentary DNA in comparison to other groups. It is possible that DNA in "shelled" organisms like the foraminifera is more likely to be preserved. Such selective preservation could alter the pattern of community structure among taxonomic groups with different cell architecture, but the observation from foraminifera makes to hypothesize that as long as the preservation pattern of DNA *within* a given taxonomic group remains similar, the eDNA of such group should conserve its biogeographic and community structure. If this hypothesis could be confirmed, this, together with the latest developments in sequencing technologies, would open new avenues for paleoceanography and paleoecology, including the investigation of the impact of major past climate crises on oceanic communities, and the genetic detection of organisms not preserved in the fossil record. This is extremely important, now that the Tara-Oceans global metabarcoding survey has shown that the largest portion of plankton biodiversity is composed of heterotrophic protists, parasites and symbionts that do not fossilize (de Vargas et al., 2015). In these regards, the information potentially preserved in deep sea sedimentary *a*DNA will likely revolutionize our understanding of the past ecology of marine plankton.

**6 Data availability**

Raw sequence data generated by Lecroq et al. (2010) and used in the present study are registered at the NCBI's Short Read Archive under the BioProject number PRJEB2682.

**7 Supplements**

Supplement 1

Detailed information on environmental samples and sequence data.

Supplement 2

Fasta files of the reference database and of the e-ribotypes attributed to planktonic foraminifera.

Supplement 3

Occurrences and taxonomic assignation of the e-ribotypes. Field explanations are given in the file.

Supplement 4

Individual alignments, phylogenetic trees and interpretation used to cluster the e-ribotype into genotypes.

## 7 Author contribution

R.M., F.L., M.K., L.O, J.P., S.M., and C.d.V conceived the project. B. L-B generated the original data. R.M., K. F. D., C.d.V. and M.K generated the reference database. F.L, R.M and M.K analyzed the dataset. R.M., F.L and M. K. wrote the manuscript and M.P., C.d.V., L.O., J.P., S.M., K.F.D., B. L-B., provided critical discussions and editions to the manuscript.

## 8 Competing interests

The authors declare no competing interests.

## 9 Acknowledgements

The study was supported by Swiss National Science Foundation grants 31003A-140766 and 313003A-159709 and by the DFG-Research Center/Cluster of Excellence "The Ocean in the Earth System".

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

**Figures**

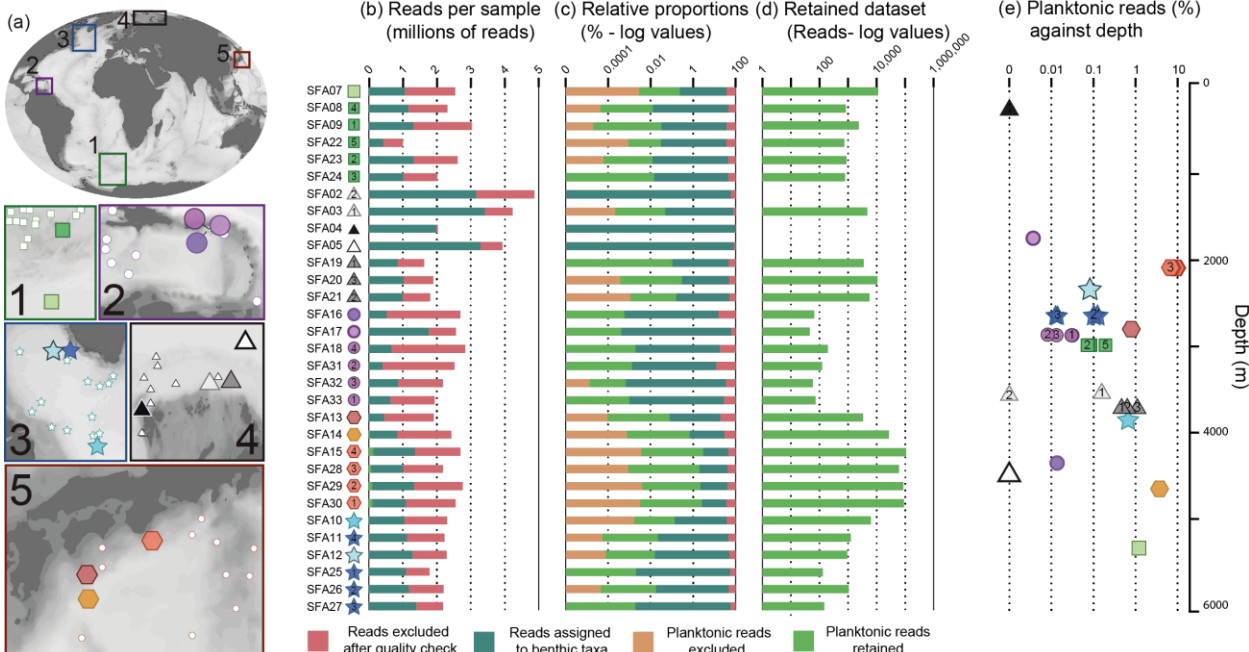

**Figure 1.** Occurrences of planktonic foraminifera in abyssal sedimentary *e*DNA. (a) Geographic location of the samples. The boxes indicate the location the core top samples in the 5 sampled regions. The larger symbols indicate the location of the samples used for *e*DNA analysis generated by Lecroq et al. (2011) and the smaller open symbol the location of the census count from the MARGO database (Kucera et al., 2005). (b-d) Results of the filtering and assignation of the dataset. The symbols with numbers correspond to the replicates of a single location shown on (a), next to the libraries name. The replicates are subsamples originating from the same gear (Lecroq et al., (2010)). (e) Relative proportions of planktonic reads in the individual samples in logged values plotted against depth.

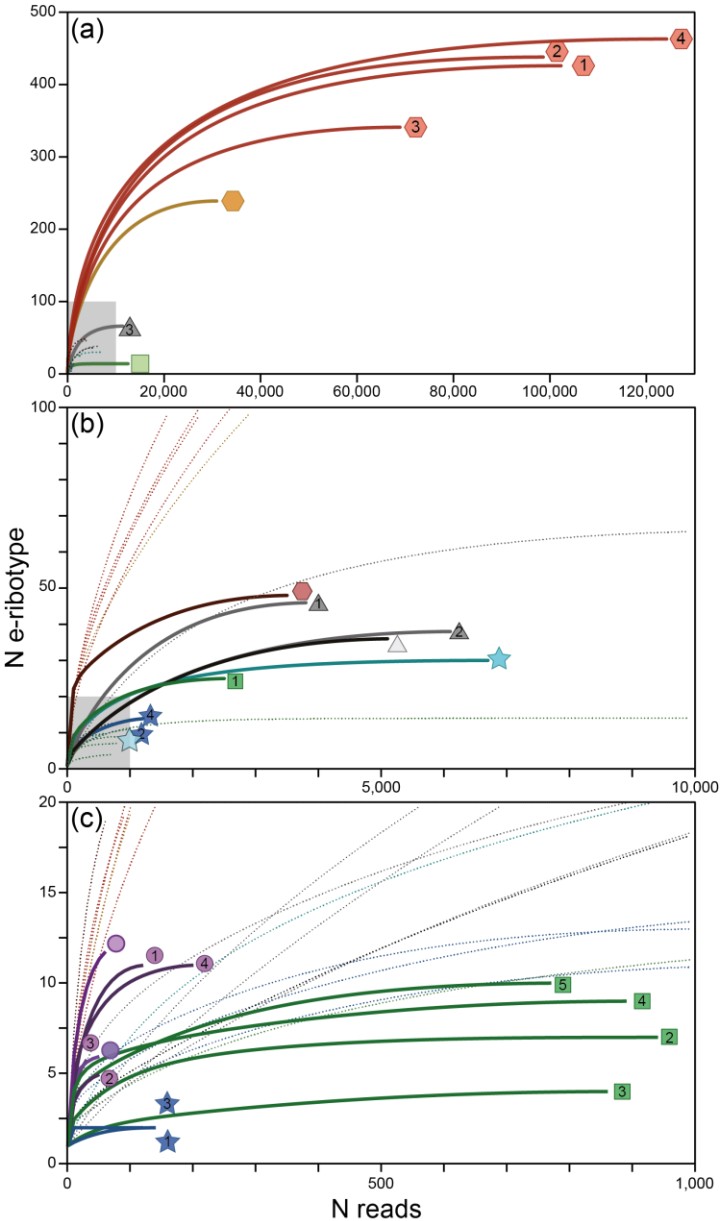

**Figure 2.** Rarefaction curves. e-ribotypes rarefaction curves of each of the 28 samples containing planktonic foraminifera sequences. The three boxes show the same rarefaction curves at 3 different scales highlighted by grey rectangles. ame symbols as. For each magnification, the curves which are out of range are drawn in dashed lines to ease the reading to the figure.

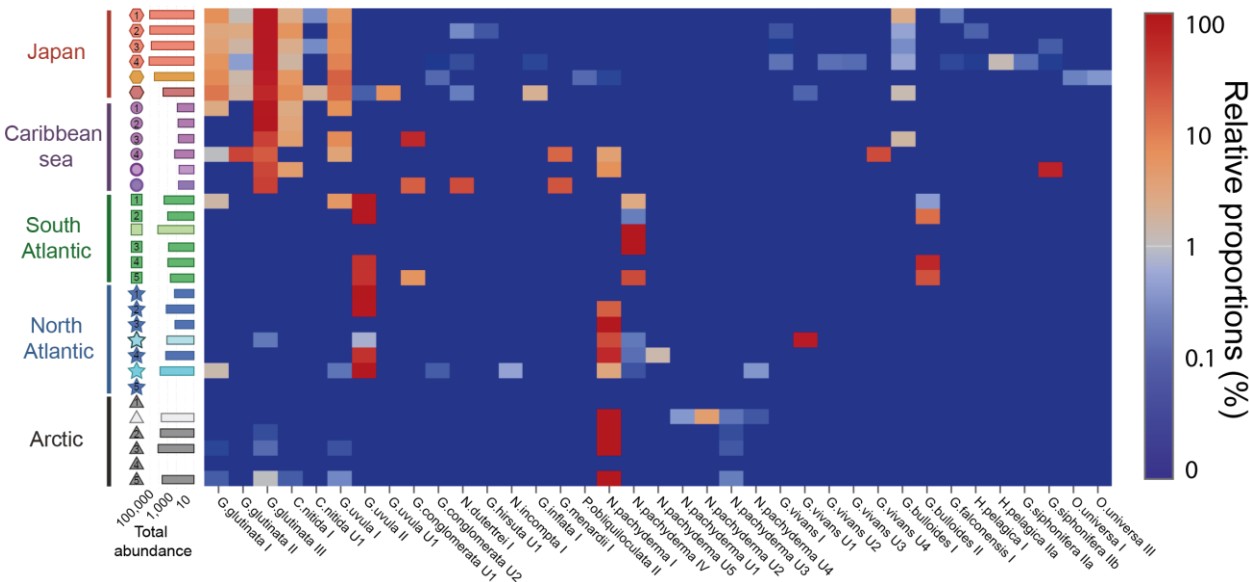

**Figure 3.** Heat Map of the relative proportions of the genetic types detected in the 31 samples. The histogram on the left side of the heat map indicates the total abundance in log-value of the reads belonging to planktonic foraminifera. Symbols as on Figs 1-2.

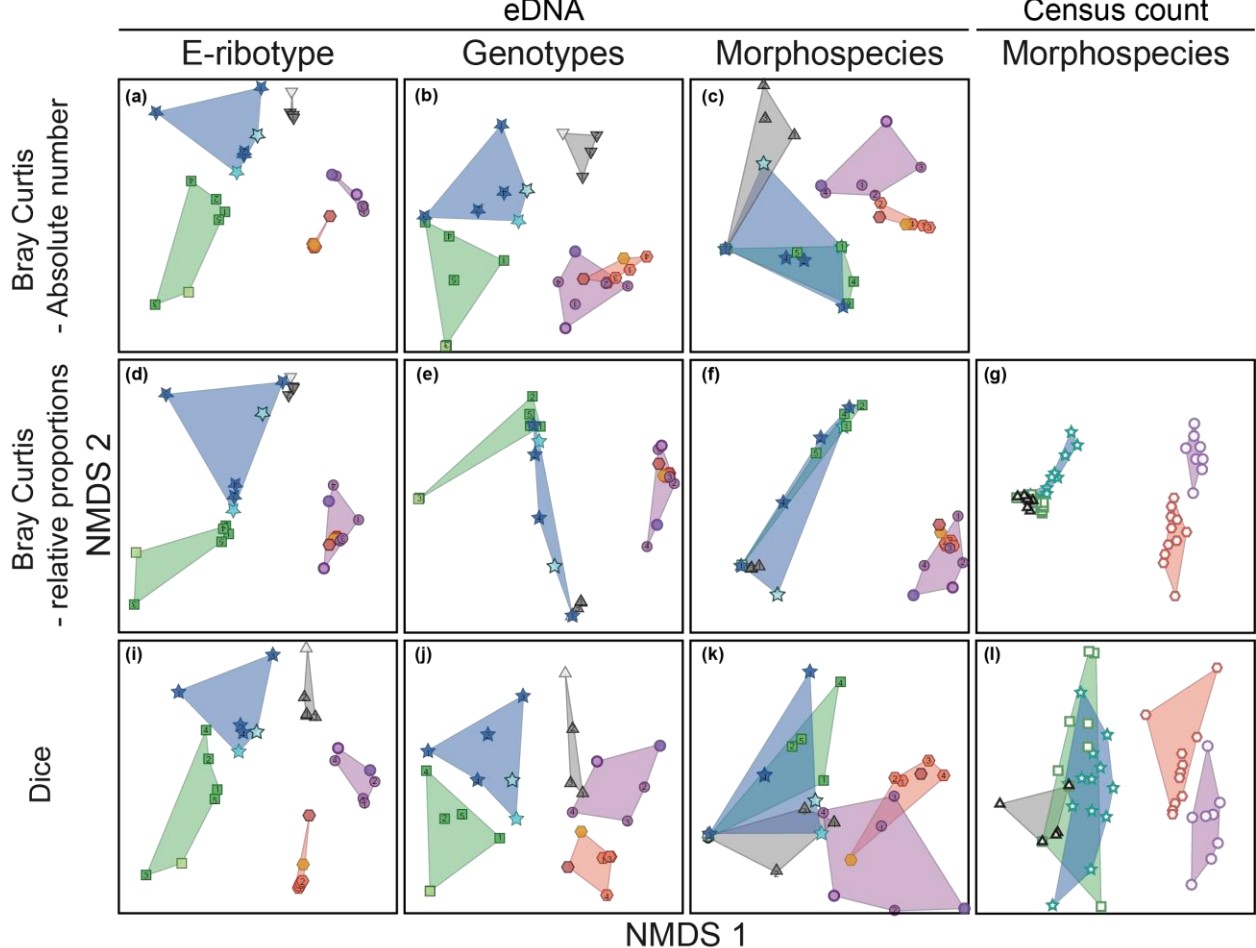

**Figure 4.** Community structuring of planktonic foraminifera in sedimentary *e*DNA. Grouping of *e*DNA and census count samples according to their taxonomic composition using Non-linear Multi-Dimensional Scaling based on Bray-Curtis (Absolute number (a-c) and relative abundances (d-g)) and Dice distances (i-l) based on corrected data. The NMDS are provided for the three different degrees of taxonomic resolution (ribotypes, genotypes and morphospecies) for the *e*DNA samples. As the census count are relative abundances, the Bray-Curtis on absolute value is not provided for the census count assemblages. The area covered by the samples of each region is highlighted. Symbols as in Fig. 1b and Fig. 2.

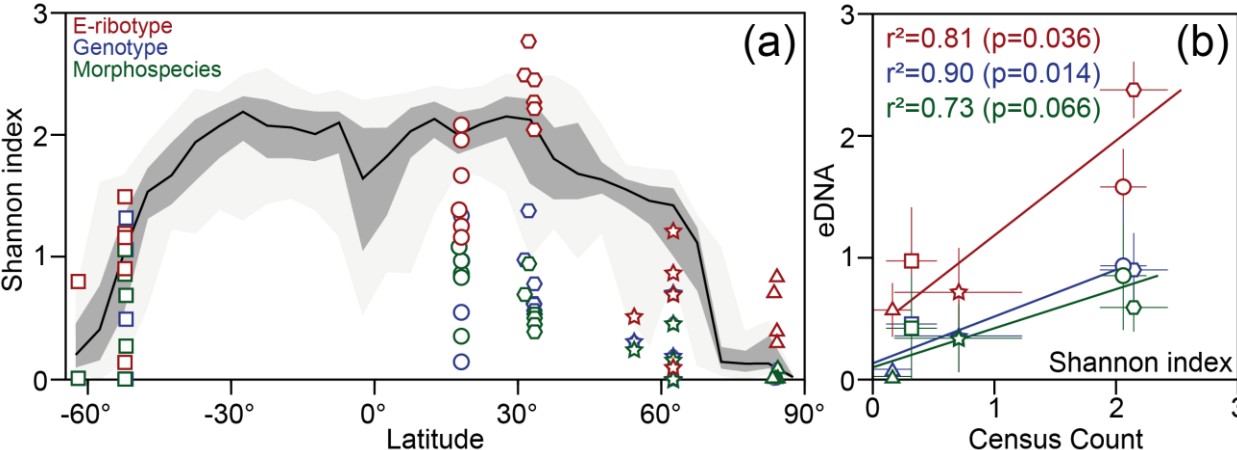

**Figure 5. Macro ecological pattern of spatial diversity known as the latitudinal gradient of diversity**. **(a)** The grey areas represent the distribution of the Shannon index calculated on the census count of planktonic foraminifera in core top samples from the MARGO database (Kucera et al., 2005) against latitude. The dark grey area represents the 1st-3rd quartiles (50% confidence interval), light grey the 5th-95th percentile (90 % confidence interval), and the black line is the median. The same similarity measure has been calculated at each location for the *e*DNA samples based on the relative abundances with the three levels of taxonomic resolution. **(b)** Relationships between mean Shannon index calculated at regional levels (symbols as on Figure 1) for census count and *e*DNA assemblages, vertical and horizontal lines indicate the standard deviation. Coefficient of correlation and p values are provided for the three taxonomic levels but are only indicative because the number of data point is too low to draw definitive conclusion.

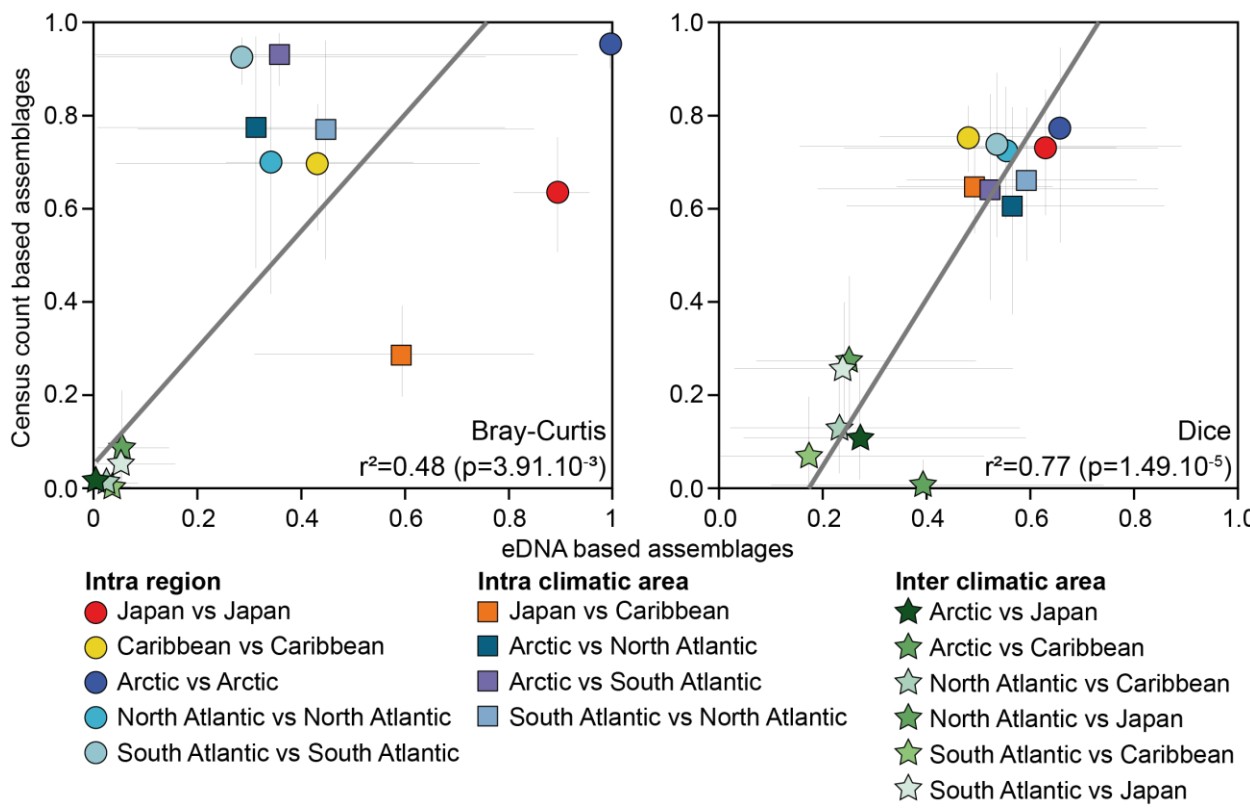

**Figure 6. *e*DNA vs census counts**. Similarity pairwise comparison of the community composition inferred from relative abundances of morphospecies based on *e*DNA and census counts among and between the five sampled regions based on the Bray-Curtis and Dice indices. Each symbol corresponds to the average between all pairwise distances of each category and the lines represent the standard deviation. The gray lines represent the linear regression, with r² and p values provided in the right bottom corner of each graph.

**Table**

**Table 1.** Sequential Bonferonni significance p-values of one-way PERMANOVAs tests associated to NMDS (Fig .4) for pairwise comparisons of regions for each taxonomic resolution and indices. Significant values (p<0.05) indicating that two regions have different distribution are shown in bold.

| | Taxonomic levels | | |
|---|---|---|---|
| Pairwise comparisons | E-ribotype | Genotypes | Morphospecies |
| *Bray-Curtis - Absolute numbers* | | | |
| Japan - Caribbean sea | **0,0021** | **0,0356** | **0,0052** |
| Japan - South Atlantic | **0,0024** | **0,0024** | **0,0018** |
| Japan - North Atlantic | **0,0028** | **0,0019** | **0,0027** |
| Japan - Arctic | **0,0039** | **0,0057** | **0,0027** |
| Caribbean sea -South Atlantic | **0,0020** | **0,0024** | **0,0024** |
| Caribbean sea -North Atlantic | **0,0026** | **0,0020** | **0,0017** |
| Caribbean sea - Arctic | **0,0048** | **0,0040** | **0,0052** |
| South Atlantic - North Atlantic | **0,0026** | **0,0040** | 0,8314 |
| South Atlantic - Arctic | **0,0045** | **0,0045** | **0,0136** |
| North Atlantic - Arctic | **0,0100** | **0,0040** | **0,0144** |
| | | | |
| *Bray-Curtis - relative proportions* | | | |
| Japan - Caribbean sea | 0,3641 | 0,1409 | 0,1139 |
| Japan - South Atlantic | **0,0025** | **0,0016** | **0,0024** |
| Japan - North Atlantic | **0,0022** | **0,0021** | **0,0024** |
| Japan - Arctic | **0,0047** | **0,0043** | **0,0051** |
| Caribbean sea -South Atlantic | **0,0023** | **0,0024** | **0,0018** |
| Caribbean sea -North Atlantic | **0,0014** | **0,0022** | **0,0025** |
| Caribbean sea - Arctic | **0,0042** | **0,0058** | **0,0035** |
| South Atlantic - North Atlantic | **0,0027** | **0,0269** | 0,7729 |
| South Atlantic - Arctic | **0,0043** | **0,0042** | 0,0978 |
| North Atlantic - Arctic | **0,0082** | **0,0304** | 0,0959 |
| | | | |
| *Dice* | | | |
| Japan - Caribbean sea | **0,0026** | **0,0015** | **0,0355** |
| Japan - South Atlantic | **0,0028** | **0,0017** | **0,0021** |
| Japan - North Atlantic | **0,0018** | **0,0022** | **0,0022** |
| Japan - Arctic | **0,0049** | **0,0051** | **0,0036** |
| Caribbean sea -South Atlantic | **0,0017** | **0,0018** | **0,0017** |
| Caribbean sea -North Atlantic | **0,0012** | **0,0024** | **0,0045** |
| Caribbean sea - Arctic | **0,0052** | **0,0185** | **0,0141** |
| South Atlantic - North Atlantic | **0,0048** | **0,0061** | 0,9155 |

| | | | |
|---|---|---|---|
| South Atlantic - Arctic | **0,0042** | **0,0048** | 0,2515 |
| North Atlantic - Arctic | **0,0088** | **0,0046** | 0,3463 |