# Peer review of "Planktonic foraminifera-derived environmental DNA extracted from abyssal sediments preserves patterns of plankton macroecology"

_Biogeosciences, 2016_

## Referee Comment (RC1) · Anonymous Referee #1 · 24 Dec 2016

The manuscript entitled "Plankton-derived environmental DNA extracted from abyssal sediments preserves patterns of plankton macroecology" address relevant scientific question within the scope of BG. The manuscript focuses on the identification of planktonic foraminifera from marine surface sedimentary DNA using a metabarcoding approach. The manuscript uses already available data and focuses on a data subset from data in which the entire foraminifera diversity(including benthic taxa) was analysed. The authors are specifically interested in the planktonic taxa, because this offer the opportunity to investigate imprints of the ocean surface biota and its processes of transport and deposition of planktonic eDNA to the oceans bottom.

I like the idea of the manuscript, because there are especially in the marine realm only

a few papers that focus on sedimentary DNA. Especially in deep ocean sediments it is a valid question to ask how much eDNA from planktonic organisms reaches the bottom and which is then also usable as an archive for planktonic taxa diversity changes.

As the author have specific knowledge about the planktonic taxa and a well-documented reference database, the detection of specific and rare taxa seems to be possible and valid. However, I have the feeling that the small subset of the entire data might be problematic in terms of the complete diversity of planktonic taxa. I have the feeling that the dominance of benthic taxa in their re-analysed data set is caused by the effect that DNA from benthic organisms is less degraded and is therefore preferentially amplified by PCR. By this the reads are dominated by benthic taxa, which reduces the number of planktonic sequences. I would be interested to see next to the relative proportion also the absolute number of sequence reads of benthic compare to planktonic reads. If you have strong variations between the total read numbers of the samples, this might be the reason that in some cases you detected only a very small fraction of planktonic reads, which is then not representative for the sample in terms of abundance and diversity. I think this makes it difficult to see global trends in planktonic foraminifera taxa in this dataset. Further I would suggest that the author show that the genetic marker used is equally specific to benthic as well as planktonic taxa.

Please see more comments in the pdf attached.

Please also note the supplement to this comment:
http://www.biogeosciences-discuss.net/bg-2016-486/bg-2016-486-RC1-supplement.pdf

**Supplement:**

[revised manuscript text omitted]

---

## Referee Comment (RC2) · Anonymous Referee #2 · 10 Jan 2017

This paper explores the potential of using eDNA from sediments to infer plankton community structure. The authors analyze already available foraminifera sequence data to address an ecological relevant and timely question. The idea of using the aDNA preserved in sediments as an archive to explore planktonic biogeographic patterns is interesting; however, as the authors focus only on planktonic foraminifera, I do not think that the authors can conclude that the approach is valid for all planktonic taxa, and the title and some conclusions should then be accordingly revised. Overall, the manuscript is well written and the methodology description is detailed and precise, although there are some important concerns that should be clarified or revised before publication. Furthermore, the results do not so strongly support the main conclusion,

as eDNA and fossil record only produce somewhat similar patterns. The only clear reproducible pattern is the separation between high and low latitude samples. A major issue relates to the large variation in the number of retained reads per sample after filtering (48 to 124,355). The authors conduct all the analyses without subsampling to the lowest number of reads, which undoubtedly bias community comparisons and diversity estimates. As some samples have an extremely low number of reads belonging to planktonic foraminifera, I suggest excluding those samples and reanalyze the dataset equalizing the number of reads per sample. In addition, the authors should clarify how they analyzed data from census counts of microfossils (details of sampling, number of individuals per sample, normalization, etc.) as they use these data to validate the use of eDNA from sediments to infer planktonic foraminifera community structure. Another suggestion is to exclude all the sequences belonging to small (<150 microns) foraminifera from the eDNA dataset, as the census counts are not including this fraction of the community.

Specific comments

Title. I suggest revising the title as the authors focus only on planktonic foraminifera. Page 2, lines 13-17. These final statements are too strong. Page 2, lines 26-27. Please explain why high concentrations of DNA in sediments indicates that part derives from planktonic/pelagic organisms. Page 3, lines 25-26 and Page 4, lines 1-2. The authors should keep in mind throughout the manuscript that they are focusing on a taxonomic group that seems to be particularly suited for validating their hypothesis and thus extrapolating to all planktonic or even the entire spectrum of pelagic organisms is not straightforward. Page 7, lines 10-11. Please, provide details on the census counts dataset. Page 7, line 18. Change "ascribed" to "ascribe". Page 9, lines 21-23. I do not see that the eDNA dataset reproduce separation between Caribbean and Japan samples (only when using absolute numbers of reads due to several order of magnitude difference in number of reads). I do not find appropriate the analyses conducted with absolute read numbers considering the extreme differences among samples. Please

consider excluding panels a-c in figure 4. Page 10, line 1. I do not see that the patterns are identical, please revise. Page 10, lines 12-13. It looks from data in figure 6 that the correlation is not significant. Moreover, many data are well above or below the 1:1 line. Please provide p-value. Page 11, lines 4-5. As already commented, the patterns are not so consistent. Page 12, line 5. I suggest changing "discover" to "detection". Page 12, lines 5-7. Are these species present in the fossil record? Please, add a comment on that. Page 12, lines 18-25. If microperforate species are not represented in the fossil record, I suggest excluding these sequences from the eDNA archive in order to compare both datasets. Page 13, line 25. I suggest smoothing this statement.

---

## Author Comment (AC1) · 9 Feb 2017

Dear Editor,

Please find enclosed the revised version of the manuscript entitled "Planktonic foraminifera-derived environmental DNA extracted from abyssal sediments preserves patterns of plankton macroecology" modified after the comments of the reviewers. The revised version of the paper is provided as a supplement.

We are thankful to both reviewers for their constructive comments. The comments made by the reviewers were complementary which is why we decided to reply jointly to them. We have supplied additional information in the figures 1, 5, 6 and also modified

the figure 4 to make its interpretation easier. We have also rewritten part of our conclusion to nuance statements and also clarify technical points following the comments of the reviewers. We have tried to answer the issues raised by the reviewers as constructively as possible and hope that it helps to further clarify our paper. The detailed answer to the comments are below.

We believe that the modifications we have further improve the manuscript such that it could be deemed suitable for publication.

Sincerely yours,

Raphael Morard

~

In the following, the comments of the reviewers Indicated by these symbols ***...*** and our responses are indicated by these symbols »> …. «<. We specify always the line number in the modified version of the manuscript to ease the location of the corrections.

~

Reviewer #1

***The manuscript entitled "Plankton-derived environmental DNA extracted from abyssal sediments preserves patterns of plankton macroecology" address relevant scientific question within the scope of BG. The manuscript focuses on the identification of planktonic foraminifera from marine surface sedimentary DNA using a metabarcoding approach. The manuscript uses already available data and focuses on a data subset from data in which the entire foraminifera diversity (including benthic taxa) was analysed. The authors are specifically interested in the planktonic taxa, because this offer the opportunity to investigate imprints of the ocean surface biota and its processes of transport and deposition of planktonic eDNA to the oceans bottom. I like the idea of the manuscript, because there are especially in the marine realm only a few

papers that focus on sedimentary DNA. Especially in deep ocean sediments it is a valid question to ask how much eDNA from planktonic organisms reaches the bottom and which is then also usable as an archive for planktonic taxa diversity changes. As the author have specific knowledge about the planktonic taxa and a well documented reference database, the detection of specific and rare taxa seems to be possible and valid. However, I have the feeling that the small subset of the entire data might be problematic in terms of the complete diversity of planktonic taxa. I have the feeling that the dominance of benthic taxa in their re-analysed data set is caused by the effect that DNA from benthic organisms is less degraded and is therefore preferentially amplified by PCR. By this the reads are dominated by benthic taxa, which reduces the number of planktonic sequences. ***

»> The reviewer correctly highlights the fact that the majority of the amplified DNA belongs to benthic taxa. This is to be expected because only benthic foraminifera are alive on the seafloor and thus deliver intact "live" DNA. As a result, the number of reads from the environmental samples that could be assigned to plankton varied and in some samples was small. This could theoretically affect the results but only if it can be shown that the patterns we observe result from unequal number of plankton-assigned reads among the samples. We are aware of this problem and highlighted this in the discussion, page 13 on lines 1-16 in the modified version. We observe that the patterns are replicable with assemblage composition in the Caribbean samples consistent with other tropical sites despite extremely low number of reads that were recovered (see Fig. 4). The lack of specificity of the primers used for the amplification of foraminiferal DNA was because the sequence libraries were originally generated to study the distribution of benthic taxa, not to study the deposition of planktonic eDNA on the seafloor. Future studies of planktonic eDNA could use more specific primers increasing the yield of the target sequences.«<

***I would be interested to see next to the relative proportion also the absolute number of sequence reads of benthic compare to planktonic reads.***

»> We have added to the figure 1 three histograms to show (1) the absolute number of reads generated for each sample (2) the relative proportions of benthic and planktonic taxa and (3) the total number of reads retained for each sample. «<

***If you have strong variations between the total read numbers of the samples, this might be the reason that in some cases you detected only a very small fraction of planktonic reads, which is then not representative for the sample in terms of abundance and diversity. I think this makes it difficult to see global trends in planktonic foraminifera taxa in this dataset. ***

»> We are perfectly aware that in some samples the full spectrum of the diversity is not recovered. The present article does not intend to characterize the diversity of foraminifera, but rather by taking advantage of our knowledge of it, to show that the transfer of eDNA from the plankton to deep-sea sediment preserves community structure and known macroecological patterns. However, we point out that despite the strong variation in the total number of reads between Japan and the Caribbean regions (Figs. 1, 2), the dominant species remain the same (Fig. 3), which explains why the structure of both sets of communities is replicated within each of these regions (Fig. 4d). We acknowledge that we are not trying to demonstrate that the full spectrum of planktonic Foraminifera diversity is recovered in the re-analyzed dataset. We specify in the discussion section that a certain minimal sequencing effort would be necessary to recover it (page 13, lines 4-7 in the modified version).«<

*** Further I would suggest that the author show that the genetic marker used is equally specific to benthic as well as planktonic taxa. ***

»>This analysis can be easily done using the reference database for planktonic taxa by Morard et al. (2015). This was indeed the first step in our analysis to check the specificity of the marker and the results are presented in the modified manuscript on page 6 lines 13-27 and page 7 lines 1-4. Because of the generally higher substitution rates among planktonic taxa (de Vargas, C., Zaninetti, L., Hilbrecht, H. & Pawlowski,

J. (1997). Phylogeny and rates of molecular evolution of planktonic foraminifera: SSU rDNA sequences compared to the fossil record. Journal of molecular evolution 45, 285–294), we observe (and report on page 4 lines 21-25 in the modified version) that the genetic marker has a specificity among planktonic taxa which in most cases allows recognition of genetic divergence below the level of morphological species, i.e. better than for benthic taxa.«<

Detailed comments of Reviewer #1:

***Page 2, Line 11: "i would not use this term" (for the word "foreign") ***

»>We have removed the word "foreign". Page 2, line 13 in the modified version.«<

***Page 2, Line 22 "planktonic foraminifera are found in the sediment? Please give some more details. is this fraction representing the diversity of the planktonic taxa? Does taphonomy lead to the preservation of only a small fraction?" ***

»> At this stage of the introduction, we are only providing background information and we are not specifically explaining the case of planktonic foraminifera. We chose to provide the information requested by the reviewer later in the text where we present the planktonic foraminifera at the page 4, lines 11-25 in the modified version.«<

***Page 3, Lines 19-21 "I would suggest to place this at the beginning of the section, because this is a major fact and all designed barcodes and their limitations depend mostly on this fact." ***

»>We have restructured this paragraph accordingly to the reviewer's suggestion. Page 3, line 19-20 in the modified version.«<

***Page 4, Line 2. "I suggest to use allothonus rather than foreign" ***

»> We have made the change thorough the text.«<

***Page 4, Lines 9-11. "this can only be done when the barcode is phylogenetically informative, please add a comment on that." ***

»>We have detailed our argumentation in a restructured paragraph. We have specified that among the planktonic foraminifera, there are segments of the rDNA which are highly informative despite short length. The chosen barcode is one of such segments. Page 4, lines 19-25 in the modified version.«<

***Page 4, Line 20. "a dot is missing." ***

»> Dot added.«<

***Page 5, Lines 7-11 "please add a sentence if you sequenced also negative controls or did extractions blanks, if yes please add a comment on the cleanliness of the controls" ***

»> The negative controls have not been sequenced in the initial study. In order to control the cleanness of the procedure, one library consisted in the extraction of a freshwater foraminifera (So not present in the marine samples). The sequenced library consisted only of reads derived from this foraminifera and no potential contaminant were found, confirming the cleanness of the dataset. These procedural steps were explained in the study of Lecroq et al. (2011) and we refer to this study accordingly. Nonetheless, we added those details on page 6, lines 5-11 in the modified version. «<

***Page 5, Lines 14-16 "I don't understand this sentence? Do you mean rank as a taxonomic unit? Please clarify"***

»> We have provided additional information to explain the organization of the database. We have specified that the classification scheme is organized hierarchically, just like classical taxonomy. Page 6, lines 15-23 in the modified document.«<

***Page 5, Lines 19-24 "how do you differentiate the genetic types? How do you differentiate genetic types from cryptic types? Do you use a threshold/cut-off (% of sequence similarity) to identify cryptic types and group them to known sequences? How do you differentiate authentic types from possible sequencing or PCR errors?" ***

»> We here reply successively to the answers of the reviewer: - We differentiate the

genetic types in the plankton using phylogenetic inferences and/or automated delimitation methods to delineate clusters of sequences which are then compared to biogeographic and ecological data (See Morard et al. (2015) and references herein). We provide those details on Page 6, lines 19-23 of the modified document. - Genetic types or cryptic species are equivalent. We specify it on page 6 line 27 of the modified document. - We use a phylogenetic approach to group sequences into genetic types in the present study (Page 8, Lines 1-16 in the modified documents) to aggregates the e-ribotypes into cryptic species. We use a phylogenetic approach because no unique threshold exists to delineate cryptic species in planktonic foraminifera (Morard et al., 2016). In doing this, we aggregate variants (genuine and potential artefacts) into single units.«<

***Page 6, Lines 11-13. "an abundance of ten only occurring once in a dataset is a very small number. I would suggest that this is only valid for sequence types which have a 100% match to your well-curated references. I think it is always more important that sequence types occur indepently in two PCR reactions. Do you mean by samples your sediment samples or also the replicates of the PCR?" ***

»> The majority of the sequence motives we analyzed were found in multiple samples and thus were generated by independent PCR. The referee is right to point out that we also include in our analysis sequences that only occur in one PCR product (albeit at least 10 times). These sequences collectively represent 7,6% of the diversity and 1,17 % of the volume of the analyzed dataset. It would have indeed been advisable to use only sequences that were replicated among independent PCR reactions. However, our sampling coverage is far from global so that it is conceivable that some sequences were originally indeed only present in one sample. This is why we decided to keep them in the analysis. We believe the retained sequences are unlikely to represent sequencing errors, because they were retained due to high similarity to the reference database. The referee agrees that this is a valid approach but indicates that a more safe approach would be to use only motives with 100% similarity. This is certainly correct and would be

important if the purpose of the study would be to describe or characterize the patterns of eDNA. However, we only use the data to test if the patterns are congruent with large scale plankton macroecology. In future studies, the best approach would be to increase the level of replication and only retain sequences occurring in independent PCR reactions. Since the production of the dataset we use, sequencing costs have dropped hugely and such replication is now a standard.«<

***Page 7, Lines 20-24. "13 genotypes were then new genotypes, which were detected by phylogenetic clustering? What do you mean by known genotypes? please clarify" ***

»>Known genotypes as those that have been described from material obtained from single-cell DNA extraction on living foraminifera. We have rewritten this section to make it clearer. Page 9, lines 8-10 in the modified version.«<

***Page 8, Line 5. "please provide the absolute numbers" ***

»>We have added histograms to the figure 1 to show the absolute numbers of reads obtained for each sample. «<

***Page 8, Lines 11-15. "I would rather write that the abundance of planktonic reads seemed not to be related to the latitutde or depth of the sample location, as planktonic types were found at different locations and in different depth showing no significant trends. I also think that you might need more sample localities to investigate such questions at global scale. " ***

»>We have rewritten this section, however we kept the interpretation that the relative amount of planktonic eDNA does not seem to decrease with depth, because it represents a specific hypothesis on DNA preservation (less DNA preserved at deeper sites due to longer time for degradation in the water column during sinking). Page 9, lines 19-26 in the modified version.«<

***Page 10, Lines 1-2. "could you please clarify: what do you mean with fossil record,

you mean morphologically identified taxa? How can it be "identical" and "with only a partial coverage" at the same time? that means both records eDNA and morphology do not represent or only partially represent the real planktonic diversity?" ***

»>To avoid any confusion, we have changed "identical" to "similar" in the paper. The fossil record refers here to the species identified by morphology. We have also better explained census counts at the end of the method section after a request of the second reviewer. Page 8, lines 21-23 in the modified document.«<

***Page 10, Line 8. "please explain, what do you mean with three taxonomic levels? I think you mean genotype, ribotype and morphospecies, but in my opinion this are no taxonomic levels. As all of the three can be assigned e.g. to species level. This are rather taxonomic units or groups." ***

»> In this particular sentence we highlight the fact that the diversity gradient is present among taxa at all three hierarchical levels. Page 11, line 24 in the modified version.«<

***Page 11, Lines 19-21. "it is not only a matter of abundance but also a matter of primer specificity to the benthic and planktonic species. Please comment on this. Please provide more information on the primer specificity between benthic and plank-tonic taxa." ***

»> This discussion is important and is present later in the manuscript. At this point in the text, we prefer to restrict our discussion to the variation of planktonic reads among the samples, which are equally affected by the primer biases. However, we then ex-pand on the impact of the primer biases later in the text p13, lines 3 to 18 in the modified version.«<

***Page 12, Lines 5-7 "they were also not identified via phylogenetic analyses? Are these species only well-represented in the database or are they also abundant as fos-sils in your data set?" ***

»> These species are both abundant in the fossil record and well represented in our

database. we have rewritten this part of the text to make it clearer. Page 14, line 4 in the modified version.«<

***Page 12, Line 24 "if you propose primer mismatch as the major bias in your analyses, could you please provide a short notice on the specificity of the primer. Please check for possible annealing preferences of your primers. I think this could be easily done, if you have a well-curated database. Software like ecoPCR could be helpful for such a test see Ficetola et al 2010" ***

»> In the case of foraminifera, the number of sequences affected is small so we could carry out this analysis manually. The number of mismatches between the common species and the primer (up to 5 mismatches for one of the two primers for each of the two species) and is reported on page 14, lines 8-10 in the modified version.«<

***Page 13, Lines 19-21 "Do you argue specifically for foraminifera records or generally for ancient sedimentary DNA? Maybe you should include to very recent papers on marine sedimentary DNA in your discussion: Solomon et al. 2016 DOI 10.1007/s13205-016-0482-y and Kirkpatrick et al. 2016 doi:10.1130/G37933.1"***

»>Here we meant ancient sedimentary record in general. We have slightly modified the first sentence of the conclusion to clarify that we talk about the ancient sedimentary DNA in general (Page 15, line 15 in the modified document). We have integrated the study of Kirkpatrick et al. (2016) in the introduction of the paper (Page 3, lines 12-13 in the modified document). The paper of Solomon et al. (2016) is about enriching sediment sample in bacteria to further isolate their DNA. This has nothing to do with our work, therefore we did not integrated it. «<

***Page 13, Line 22. "what do you mean with taxonomic bias? different preservation dependent on different taxa? Primer bias...?Please clarify." ***

»> Here we mean preferential preservation that would induce taxonomic bias. We now specify it in the sentence page 15 line 23 in the modified version.«<

***Page 21 line 5. "Does this mean that not all samples were replicated? What was the reason for this? " ***

This comment is directly related to the following comment and both are addressed together:

*** "do the replicates show also similarities in the genetic types obtained? Or are there large differences? Are these replicates PCR replicates of the same DNA extract? " ***

»> Not all the samples have been replicated but we have to keep in mind that these data are now more than 6 years old and that the standards in metabarcoding have drastically evolved since then. At the time these data were generated, the access to sequencing technology was limited and the costs prohibitive, limiting the possibility of full replication, forcing the authors to make choices between replication and sequencing depth. The replicates are DNA isolates extracted from subsamples of sediments recovered from the same deployment (now quoted in the caption of the figure). Discrepancies can exist between replicates (see Lejzerowicz, F., Esling, P. & Pawlowski, J. (2014). Patchiness of deep-sea benthic Foraminifera across the Southern Ocean: Insights from high-throughput DNA sequencing. Deep Sea Research Part II: Topical Studies in Oceanography. Elsevier 108, 17–26.). However these are lower than the inter region comparisons. Page 23, lines 3-11.«<

***Page 23 line1. "what do you mean with geneotypes here? Do you refer to e-ribotypes or genetic types?" ***

»>Here genotypes means genetic types. We have made the change in the text to avoid confusion. Page 25, line 1 in the modified version. «<

***Page 25, line 2. "Please explain the mid-domain effect in the manuscript, it has never been mentioned before." ***

»>This was an oversight on our side. We have replaced "mid-domain effect" by "latitudinal gradient of diversity" which is used in the text and appropriate to describe the

pattern. Page 27, line 4 in the modified version. «<

***Page 28, line 2. "verstehe ich nicht? dass die grünen clustern".***

»>The green stars are clustering because they represent comparisons of sampels from different climatic regions, where the communities are dissimilar, and thus have a low score in pairwise comparison, both for fossil and eDNA assemblages. Page 28.«<

Reviewer #2

***This paper explores the potential of using eDNA from sediments to infer plankton community structure. The authors analyze already available foraminifera sequence data to address an ecological relevant and timely question. The idea of using the aDNA preserved in sediments as an archive to explore planktonic biogeographic patterns is interesting; however, as the authors focus only on planktonic foraminifera, I do not think that the authors can conclude that the approach is valid for all planktonic taxa, and the title and some conclusions should then be accordingly revised. ***

»> The only way to prove that planktonic signature is preserved in sediments is to use a group for which fossil record exists, a curated reference database is available and where a short barcode have enough resolution to allow assignation of environmental read at the species level for direct comparison. To our knowledge, planktonic foraminifera is the only group meeting these criteria. In our opinion we do not conclude in the paper that the results we observe apply to all plankton. We avertedly used the word "hypothesize" in the conclusion to propose that if the conservation of the ecological patterns is true for foraminifera it should be the case for the rest of the pelagic community because we have no reason to believe that the transfer of organic matter from the top to the bottom of the ocean acts differently depending on taxonomic group. We hope that our study will motivate other studies to exploit this potential source of information still locked into the sedimentary archive.«<

***Overall, the manuscript is well written and the methodology description is detailed

and precise, although there are some important concerns that should be clarified or revised before publication. Furthermore, the results do not so strongly support the main conclusion, as eDNA and fossil record only produce somewhat similar patterns. The only clear reproducible pattern is the separation between high and low latitude samples. A major issue relates to the large variation in the number of retained reads per sample after filtering (48 to 124,355). \*\*\*

»> We have discussed this point in our paper (Page 13, line 3-18 in the modified version). It is actually remarkable that despite the large variation in the number of reads among samples, it is still possible to recover the structure of the community. The Caribbean and Japan samples have respectively the lowest and highest amounts of reads on average, and are found clustered together when comparing the relative proportion of their community, regardless to the taxonomic level that is considered (Fig. 4d-f). In addition, there are clear patterns which are reproducible at the regional scale, which is a strong result since it shows that the preservation of planktonic eDNA in sediments is not random.«»

\*\*\*The authors conduct all the analyses without subsampling to the lowest number of reads, which undoubtedly bias community comparisons and diversity estimates. \*\*\*

»>We have considered all reads in our primary analysis of the dataset, and these data are available in the supplementary information 2. We have not identified a higher diversity in the reads in low abundance compared to the dataset we retained. We have concluded that those reads were variants of abundant reads rather than genuine sequences. Their inclusion in the final dataset would have rendered its analysis more difficult, but not more complete. We point out that reviewer #1 made the exact opposite comment on our strategy, that retaining e-ribotype with an abundance of 10 only is a too low number. This signifies that there is no clear consensus on the strategy to follow and that the strategy we chose is a balanced compromise between two opposite philosophies. «<

***As some samples have an extremely low number of reads belonging to planktonic foraminifera, I suggest excluding those samples and reanalyze the dataset equalizing the number of reads per sample. ***

»> This is an important issue and we have therefore considered the data from several points of view, successively reducing the effect of sequencing depth. We have done an analysis including the total number of reads recovered, then analysed the relative proportion of the reads only, which removes the information on read number and finally an analysis of diversity which removes the information on read number in a different way. These analyses are showed in figure 4, where we have performed NMDS successively on the dataset using Bray-curtis indices on the absolute number and relative proportions, and finally the Dice indices that consider only presence/absence data. As said above, it is remarkable that despite the low number of reads retrieved in some samples it was possible to retrieve the community structure (relative proportions). We have no reason to believe that re-analyzing the dataset by doing another form of equalization of sequencing depth would bring a different conclusion. «<

***In addition, the authors should clarify how they analyzed data from census counts of microfossils (details of sampling, number of individuals per sample, normalization, etc.) as they use these data to validate the use of eDNA from sediments to infer planktonic foraminifera community structure. ***

»> We have provided further information to contextualize the data on census counts. Page 8, lines 21-23 in the modified document «<

***Another suggestion is to exclude all the sequences belonging to small (<150 microns) foraminifera from the eDNA dataset, as the census counts are not including this fraction of the community. ***

»> Some microperforate species also occur in the larger fraction, but in different proportions than in the small fraction (Brummer et al. (1986), quoted in the text). Removing them from the data would in our opinion decrease the number of taxa in the analyses

too much. »<

Specific comments

\*\*\*Title. I suggest revising the title as the authors focus only on planktonic foraminifera. \*\*\*

»>Whilst we maintain that it is extremely unlikely that the results we obtained apply to the plankton at large, we have to acknowledge that the results are only based on planktonic foraminifera and so we have revised the title accordingly. «<

\*\*\*Page 2, lines 13-17. These final statements are too strong. \*\*\*

»> We have hedged these statements along the lines described above with regard to the likelihood that the results we obtain apply to plankton at large. Page 2, lines 16-21 «<

\*\*\*Page 2, lines 26-27. Please explain why high concentrations of DNA in sediments indicates that part derives from planktonic/pelagic organisms. \*\*\*

»> Here, we are just stating that the vast majority of sedimentary DNA is extracellular, we do not say that this DNA originate from plankton. Which is why we state in the next sentence that the DNA "survives" even after the death of an organism, therefore at least a part of this extracellular pool derives from the organisms inhabiting the water column. In addition there are good evidences that part of the DNA present in the abyss derive from phototrophic taxa that occur in plankton only (Pawlowski, J., Christen, R., Lecroq, B., Bachar, D., Shahbazkia, H. R., Amaral-Zettler, L. & Guillou, L. (2011). Eukaryotic richness in the abyss: Insights from pyrotag sequencing. PLoS ONE 6). «<

\*\*\*Page 3, lines 25-26 and Page 4, lines 1-2. The authors should keep in mind throughout the manuscript that they are focusing on a taxonomic group that seems to be particularly suited for validating their hypothesis and thus extrapolating to all planktonic or even the entire spectrum of pelagic organisms is not straightforward. \*\*\*

»>As said in the reply to the general comment, we use foraminifera only to demonstrate that the imprint of planktonic DNA is preserved in sediments. We only hypothesize that if it is true for planktonic foraminifera, it should be the same for other organisms that leave no fossil record.«<

***Page 7, lines 10-11. Please, provide details on the census counts dataset. ***

»>We have provided further details. Page 8, lines 21-23 of the modified document.«<

***Page 7, line 18. Change "ascribed" to "ascribe". ***

»>We have made the change. Page 9, line 2 in the modified document.«<

***Page 9, lines 21-23. I do not see that the eDNA dataset reproduce separation between Caribbean and Japan samples (only when using absolute numbers of reads due to several order of magnitude difference in number of reads). I do not find appropriate the analyses conducted with absolute read numbers considering the extreme differences among samples. Please consider excluding panels a-c in figure 4. ***

»> We disagree with the interpretation of the figure 4 by the reviewer. We have modified the figure 4 to make the pattern more obvious. We have reduced the size of the data points and colored the area they are covering (See the modified version of the manuscript page 26). The Caribbean and Japan areas (red and purple) are perfectly disjoint with the Dice index and using E-ribotype (Fig. 4i) and Genotypes (Fig. 4j) taxonomic levels. They are also disjoint with the Bray-Curtis index when using the absolute number (Fig. 4a and 4b). It is true that they are partially overlapping at the morphospecies level, but it is caused by a single sample. We consider that the panel a-c are useful because they show that the patterns observed are reproducible at the regional level when considering the "raw data" and this is one of the messages of our paper.«<

***Page 10, line 1. I do not see that the patterns are identical, please revise. ***

»>We have replaced "identical" with "similar" to moderate our statement. Page 11, line 21 in the modified version.«<

***Page 10, lines 12-13. It looks from data in figure 6 that the correlation is not significant. Moreover, many data are well above or below the 1:1 line. Please provide p-value. ***

»> We have added the $r^2$ and p value on the figure 6. The p value is highly significant in both cases. We have also added the p value on the correlation shown on the figure 5 to be consistent. The reviewer is right that even if the correlations are statistically supported for both graph, the relationships are not 1:1.«<

***Page 11, lines 4-5. As already commented, the patterns are not so consistent. ***

»>As mentioned above, we prefer to stick with our interpretation of the dataset and we consider that our evidences are strong enough to support the statement. «<

***Page 12, line 5. I suggest changing "discover" to "detection". ***

»>We have made the change. Page 14, line 3 in the modified version.«<

***Page 12, lines 5-7. Are these species present in the fossil record? Please, add a comment on that. ***

»>Yes, the species we are mentioning are present in the fossil record, we have change the sentence to make it clearer. Page 14, line 4 in the modified version.«<

***Page 12, lines 18-25. If microperforate species are not represented in the fossil record, I suggest excluding these sequences from the eDNA archive in order to compare both datasets. ***

»> The microperforates are present in the fossil record but they have different relative abundance depending of the size fraction which is considered. In the large fraction, used in micropaleontology, they have a modest abundance, whilst in the small size fraction, which is rarely used, they are dominant. We provide this information in the text "Alternatively, the higher abundance of reads assigned to microperforate taxa could represent a genuine pattern, questioning the representativeness of census counts of

fossil foraminifera, which ignore specimens smaller than 150 $\mu$m (Kucera et al., 2005). Microperforate species tend to be small and are therefore disproportionately abundant in the size fraction smaller than 150 $\mu$m (Brummer et al., 1986)".«<

***Page 13, line 25. I suggest smoothing this statement. ***

»>Here we are only making a hypothesis, however we have slightly modified the statement to moderate its strength. Page 15, Lines 22-24 in the modified version.«<

Please also note the supplement to this comment:
http://www.biogeosciences-discuss.net/bg-2016-486/bg-2016-486-AC1-supplement.pdf

**Supplement:**

[revised manuscript text omitted]

---

## Referee Report (RR1)

General comments

The revised version of the manuscript, although clearer in some aspects, does not address some of the concerns. Even if the authors are convinced that the extremely different sequencing depth among samples (48 to 124,355 reads per library) does not affect their analyses and conclusions they should have made the exercise of testing such potential bias and adequately discuss the problem. This is particularly relevant considering that they are including clearly un-saturated samples (Caribbean samples). I think that the authors have to justify convincingly the inclusion of samples with such an extremely low library size (<100 reads) and clearly define which size they consider defective. I could agree that sub-sampling all the libraries to the lowest library size (once excluding those defective samples with extremely low number of reads) would imply a considerable loss of data. However, there are alternative methodologies to account for widely varying library sizes. Sequence counts can be also normalized, for example, with the r package deseq2 (Love et al. 2014). This method accounts for differential sample depth and is appropriate for normalizing high-variance data sets from high-throuput sequencing.

On the other hand, the authors must discuss in more detail why the transfer of organic matter to the deep ocean and the preservation of planktonic DNA in oceanic sediments "should apply" to other taxa, as affirmed by the authors. There are some literature to this respect that they could include. For example, Capo et al (2015) showed that Cryptophyta and Haptophyta are not well preserved in lake sediments; and Boere et al (2011) discussed the possible causes behind the variation in the level of DNA preservation among diatoms and dinoflagellates in oceanic sediments. I addition, I do not believe that "there are no have no reason to believe that the transfer of organic matter from the top to the bottom of the ocean acts differently depending on taxonomic group" as stated by the authors.

Specific comments

Figure 4. I do not see that "the Caribbean and Japan areas (red and purple) are perfectly disjoint" at the taxonomic level of morphospecies. The authors can conduct some statistical analysis to test that (e.g. PERMANOVA and ANOSIM). In addition, the separation observed based on absolute number or on Dice index mostly derives from the very different level of taxonomic saturation between Japan and Caribbean samples (figure 3), as the authors mention in the manuscript.

References

Boere, A. C., Damsté, J. S. S., Rijpstra, W. I. C., Volkman, J. K., & Coolen, M. J. (2011). Source-specific variability in post-depositional DNA preservation with potential implications for DNA based paleoecological records. Organic Geochemistry, 42(10), 1216-1225.

Capo, E., Debroas, D., Arnaud, F., & Domaizon, I. (2015). Is planktonic diversity well recorded in sedimentary DNA? Toward the reconstruction of past protistan diversity. Microbial ecology, 70(4), 865-875.

Love M, Anders S, Huber W (2014) Differential analysis of count data—the DESeq2 package.

---

## Author Response (AR2)

Dear Editor,

Please find enclosed the revised version of the manuscript entitled "Planktonic foraminifera-derived environmental DNA extracted from abyssal sediments preserves patterns of plankton macroecology" modified after the comments of the reviewer.

Following the reviewer request, we have normalized the dataset using the Cumulative Sum Scaling (Paulson et al., 2013). The normalization slightly changed the results but their interpretation remains unchanged. We have also included elements in the discussion to explain better our hypothesis about the conservation of taxonomic information during the transfer of organic matter from the top to the bottom of the ocean.

We believe that the modifications we have made further improve the manuscript such that it could be deemed suitable for publication.

Sincerely yours,

Dr. Raphael Morard

In the following, the comments of the reviewers Indicated by these symbols ***…*** and our responses are indicated by these symbols >>> …. <<<. We specify always the line number in the modified version of the manuscript with track changes that is attach at the bottom of the response.

***General comments
The revised version of the manuscript, although clearer in some aspects, does not address some of the concerns. Even if the authors are convinced that the extremely different sequencing depth among samples (48 to 124,355 reads per library) does not affect their analyses and conclusions they should have made the exercise of testing such potential bias and adequately discuss the problem. This is particularly relevant considering that they are including clearly un-saturated samples (Caribbean samples). I think that the authors have to justify convincingly the inclusion of samples with such an extremely low library size (<100 reads) and clearly define which size they consider defective.****

>>> The reviewer is right that including samples with such a low read number is generally not adequate. However, we are not using these libraries to make ecological inferences but we show that despite these low numbers, the community structure (relative proportion of taxa) is conserved. We recognize in the discussion that there is a large variation in the number of reads recovered in our dataset and that it is not possible to explain specifically why (Page 13, Lines 6-16). We recognize that a minimum sequencing effort is necessary to recover the full taxonomy spectrum, but we cannot state in the article what this effort should be, since it would be mostly, group, sample type and technology dependent.<<<

***I could agree that sub-sampling all the libraries to the lowest library size (once excluding those defective samples with extremely low number of reads) would imply a considerable loss of data. However, there are alternative methodologies to account for widely varying library sizes. Sequence counts can be also normalized, for example, with the r package deseq2 (Love et al. 2014). This method accounts for differential sample depth and is appropriate for normalizing high-variance data sets from high-throughput sequencing.***

>>> Following the reviewer request, we have made the exercise of testing the effect of the biases induced by the different size of the libraries. We have used the Cumulative Sum Scaling (Paulson et al., 2013) that corrects potential bias of technical zeroes and relative proportions, and that performs equally well than deseq2 (Costea et al .2014). However, this method does not equalize the number of reads, therefore we have maintained our analyses of the dataset using Bray-Curtis distances on absolute and relative (but corrected) abundances. The structure of the resulting NMDS stayed the same, however the samples belonging to the Caribbean and Japan region are now closer, due to the reduction of the gap in the number of reads in the corrected values. Also, we have conducted a PERMANOVA test on the coordinates of the samples in the NMDS, as suggested by the reviewer, and it supports our previous interpretation that the Caribbean and Japan samples have the same distribution when considering the relative proportions at all taxonomic levels. The high latitude samples show no differences only when considering the morphospecies level, either when considering the relative proportions or presence/absence data. In line with the suggestion of the reviewer, we are now using the corrected values for all analyses. To this end, we have redrawn the figures 5 and 6 based on the corrected data, even though changes are hardly noticeable since using the Cumulative Sum scaling did not modify significantly the structure of the dataset.<<<

***On the other hand, the authors must discuss in more detail why the transfer of organic matter to the deep ocean and the preservation of planktonic DNA in oceanic sediments "should apply" to other taxa, as affirmed by the authors. There are some literature to this respect that they could include. For example, Capo et al (2015) showed that Cryptophyta and

**Haptophyta are not well preserved in lake sediments; and Boere et al (2011) discussed the possible causes behind the variation in the level of DNA preservation among diatoms and dinoflagellates in oceanic sediments. I addition, I do not believe that "there are no have no reason to believe that the transfer of organic matter from the top to the bottom of the ocean acts differently depending on taxonomic group" as stated by the authors.\*\*\***

>>>We have realized that we have been unclear in the way we have formulated this hypothesis. We meant that the preservation pattern of DNA *within* a given taxonomic group should conserve the basic biogeographic and community structure, as for foraminifera DNA, as long as this DNA could be preserved during the transfer. However, we did not mean that all the taxonomic group will have the same degree of preservation, as we agree that "shelled" DNA may be more resistant to degradation. We thank the reviewer for spotting this inconsistency. We have reformulated this sentence and incorporated the suggested references in this place of the paper (Pages 15 line 20 to Page 16 line 3).<<<

**\*\*\*Specific comments**
**Figure 4. I do not see that "the Caribbean and Japan areas (red and purple) are perfectly disjoint" at the taxonomic level of morphospecies. The authors can conduct some statistical analysis to test that (e.g. PERMANOVA and ANOSIM). In addition, the separation observed based on absolute number or on Dice index mostly derives from the very different level of taxonomic saturation between Japan and Caribbean samples (figure 3), as the authors mention in the manuscript.\*\*\***

>>> The suggested analysis has now been carried out (see above). <<<

**\*\*\*References**
**Boere, A. C., Damsté, J. S. S., Rijpstra, W. I. C., Volkman, J. K., & Coolen, M. J. (2011). Source-specific variability in post-depositional DNA preservation with potential implications for DNA based paleoecological records. Organic Geochemistry, 42(10), 1216-1225.**
**Capo, E., Debroas, D., Arnaud, F., & Domaizon, I. (2015). Is planktonic diversity well recorded in sedimentary DNA? Toward the reconstruction of past protistan diversity. Microbial ecology, 70(4), 865-875.**
**Love M, Anders S, Huber W (2014) Differential analysis of count data—the DESeq2 package.\*\*\***

>>>**References**

Paulson, J. N., Stine, O. C., Bravo, H. C. and Pop, M.: Differential abundance analysis for microbial marker-gene surveys, Nat. Methods, 10(12), 1200–1202, doi:10.1038/nmeth.2658, 2013.

Costea, P., Zeller, G., Sunagawa, S. and Bork, P. A fair comparison, Nat. methods, 11(4), 359-359, doi: 10.1038/nmeth.2897, 2014.<<<

[revised manuscript text omitted]

---

## Author Response (AR3)

Dear Editor,

Please find enclosed the final version of the manuscript entitled "Planktonic foraminifera-derived environmental DNA extracted from abyssal sediments preserves patterns of plankton macroecology".

We have made the corrections suggested by the reviewer but we chose to keep the Table 1 because it is more exhaustive to keep it and providing the p-values in the text renders the reading more difficult.

We are grateful for the comments provided throughout the entire review process.

Sincerely yours,

Dr. Raphaël Morard

In the following, the comments of the reviewers Indicated by these symbols \*\*\*…\*\*\* and our responses are indicated by these symbols >>> …. <<<. We specify always the line number in the modified version of the manuscript with track changes that is attach at the bottom of the response.

**\*\*\* Minor comments**

**-Page 2, lines 18-20. I suggest deleting the last sentence. The authors could conclude with a more general statement about the potential use of eDNA preserved in sediments. \*\*\***

>>>We have deleted the sentence and replaced it by a more general statement. Page 2, Lines 18-20.<<<

**\*\*\*-Table 1 can be omitted. I suggest just providing the PERMANOVA p-values, when required, in the text. \*\*\***

>>> We prefer here to keep the table 1 because it has been requested by one of the reviewer and it is more exhaustive. In addition, the reading of the text is easier without statistical results included in it.<<<

**\*\*\*-Page 10, line 24. I suggest deleting Table 1 citation. Also, I suggest changing "with genotype level" to "at genotype level". \*\*\***

>>> We have made the change. Page 10, line 25.<<<

**\*\*\*-Page 10, line 26. Change "genotypes" to "genotype". \*\*\***

>>>We have made the change. Page 10, line 26.<<<

**\*\*\*-Page 15, lines 17-20. Please revise the conditional use in this sentence for clarity or delete it, as it does not provide relevant information. \*\*\***

>>>We have reformulate this section of the text. Page 15, lines 9-18.<<<

[revised manuscript text omitted]